# Probing Spacetime Foam with Extragalactic Sources of High-Energy Photons

**Y. Jack Ng** [1] and **Eric S. Perlman** [2,*]

1    Department of Physics & Astronomy, University of North Carolina, Chapel Hill, NC 27599, USA; yjng@physics.unc.edu
2    Department of Aerospace, Physics and Space Sciences, Florida Institute of Technology, Melbourne, FL 32901, USA
*    Correspondence: eperlman@fit.edu

**Abstract:** Quantum fluctuations can endow spacetime with a foamy structure. In this review article, we discuss our various proposals to observationally constrain models of spacetime foam. One way is to examine if the light wave-front from a distant quasar or GRB can be noticeably distorted by spacetime-foam-induced phase incoherence. As the phase fluctuations are proportional to the distance to the source but inversely proportional to the wavelength, ultra-high energy photons (>1 TeV) from distant sources are particularly useful. We elaborate on several proposals, including the possibility of detecting spacetime foam by observing "seeing disks" in the images of distant quasars and active galactic nuclei. We also discuss the appropriate distance measure for calculating the expected angular broadening. In addition, we discuss our more recent work in which we investigate whether wave-front distortions on small scales (due to spacetime foam) can cause distant objects become undetectable because the phase fluctuations have accumulated to the point at which image formation is impossible. Another possibility that has recently become accessible is to use interferometers to observe cosmologically distant sources, thereby giving a large baseline perpendicular to the local wave vector over which the wave front could become corrugated and thus distorted, reducing or eliminating its fringe visibility. We argue that all these methods ultimately depend on the availability of ways (if any) to carry out proper averaging of contributions from different light paths from the source to the telescope.

**Keywords:** spacetime foam; quantum fluctuations; holographic principle; ultra-high energy photons; quasars; interferometry; astronomical observations

## 1. Introduction

Before last century, spacetime was regarded as nothing more than a passive and static arena in which events take place. Early last century, Einstein's general relativity changed that viewpoint and promoted spacetime to an active and dynamical entity. Nowadays, many physicists also believe that spacetime, like all matter and energy, undergoes quantum fluctuations. Thus, we expect that spacetime, probed at a small enough scale, will appear complicated—something akin in complexity to a turbulent froth that John Wheeler [1–3] dubbed "spacetime foam", also known as "quantum foam". However, how large are the fluctuations? How foamy is spacetime? Additionally, how can we detect quantum foam?

To quantify the problem, let us recall that, if spacetime indeed undergoes quantum fluctuations, there will be an intrinsic limitation to the accuracy with which one can measure a macroscopic distance, for that distance fluctuates. We denote the fluctuation of a distance $\ell$ by $\delta\ell$. On rather general grounds, we expect [4]

$$\delta\ell \gtrsim \ell^{1-\alpha}\ell_P^\alpha, \tag{1}$$

up to a numerical multiplicative factor $\sim 1$, which we will drop. Here $\ell_P = \sqrt{\hbar G / c^3}$ is the Planck length, the characteristic length scale in quantum gravity, and we have denoted the Planck constant, gravitational constant and the speed of light by $\hbar$, $G$ and $c$, respectively. (There is an analogous expression for the fluctuation of a time interval $t$: $\delta t \gtrsim t^{1-\alpha} t_P^{\alpha}$, with $t_P = \ell_P / c$ being the Planck time). We will assume that spacetime foam models can be characterized by such a form parameterized by $0 \leq \alpha \leq 1$. The standard choice [5] of $\alpha$ is $\alpha = 1$; the choice of $\alpha = 2/3$ appears [6,7] to be consistent with the holographic principle and black hole physics [8–11] and will be called the holography model [12,13]; $\alpha = 1/2$ corresponds to the random-walk model found in the literature [14,15]. Though much of our discussion below is applicable to the general case, we will use these three cases as examples of the quantum gravity models.

In this review article, we will concentrate on our work with our various collaborators to detect spacetime foam with ultra-high energy photons from extra-galactic sources. However, let us first mention some of the other ways to probe Planck-scale physics. They include: detecting spacetime foam with laser-based interferometry techniques [6,16], understanding the threshold anomalies in high energy cosmic ray and gamma ray events [17–21], looking for energy-dependent spreads in the arrival time of photons of the same energy from GRBs [22,23] and others [4].

Here is the outline of the rest of the paper: Section 2 is devoted to (1) discussing metric fluctuations, energy–momentum fluctuations and fluctuations of the speed of light as a function of energy; (2) examine the cumulative effects of spacetime fluctuations and to find out whether there can be a large-enough spread in arrival times of light from distant pulsed sources to be detectable; (3) derive the holographic spacetime foam model by considering a gedanken experiment. In Section 3, we start our discussion of using spacetime foam-induced phase scrambling of light from extra-galactic sources to probe spacetime fluctuations; we also propose to detect the texture of spacetime foam by looking for halo structures in the images of distant quasars. In the process, we argue that the correct distance to use for such measurements is the comoving distance, rather than the luminosity distance. In Section 4, we elaborate further on the proposal to detect spacetime foam by imaging distant quasars and AGNs. We show explicitly that such observations appear to rule out the holographic spacetime model. In Section 5, we discuss the recent observations of 3C 273 with the *GRAVITY* near-IR interferometer on the VLBI and future prospects for constraining spacetime foam models. We offer our summary and conclusions in Section 6.

## 2. Quantum Foam Models

In this section, we first generalize the expressions for the various fluctuations in models parameterized by $\alpha$, then examine the cumulative effects of spacetime fluctuations and finally derive the holographic spacetime foam model.

### 2.1. Fluctuations for Spacetime Foam Models Parameterized by $\alpha$

The spacetime fluctuation Equation (1) translates into a metric fluctuation over a distance $\ell$ given by $\delta g_{\mu\nu} \gtrsim (\ell_P / \ell)^{\alpha}$. Spacetime foam effects also introduce uncertainties in energy (E) and momentum (p) measurements [24,25], $\delta E \sim E (E / E_P)^{\alpha}$ and $\delta p \sim p (p / (m_p c))^{\alpha}$, respectively, where $m_P$ and $E_P = m_P c^2$ denote the Planck mass and Planck energy. These energy and momentum uncertainties modify the dispersion relation for the photons as follows:

$$E^2 \simeq c^2 p^2 + \epsilon E^2 (E / E_P)^{\alpha}, \tag{2}$$

where $\epsilon \sim \pm 1$. Thus, the speed of light $v = \partial E / \partial p$ fluctuates by $\delta v \sim 2\epsilon c (E / E_P)^{\alpha}$. Note that all the fluctuations take on a $\pm$ sign with equal probability (like a Gaussian distribution of $v$ about $c$).

## 2.2. Cumulative Effects of Spacetime Fluctuations

Consider a large distance $\ell$ (think of the distance between extragalactic sources and the telescope), and divide it into $\ell/\lambda$ equal parts, each of which has length $\lambda$ (for $\lambda$, think of the wavelength of the observed light from the distant source). If we start with $\delta\lambda$ from each part, the question is how the $\ell/\lambda$ parts add up to $\delta\ell$ for the whole distance $\ell$. In other words, we want to find the cumulative factor [12] $\mathcal{C}_\alpha$ defined by $\delta\ell = \mathcal{C}_\alpha\,\delta\lambda$. Since $\delta\ell \sim \ell_P(l/\ell_P)^{1-\alpha}$ and $\delta\lambda \sim \ell_P(\lambda/\ell_P)^{1-\alpha}$, the result is $\mathcal{C}_\alpha = (\ell/\lambda)^{1-\alpha}$; in particular, $\mathcal{C}_{\alpha=1/2} = (\ell/\lambda)^{1/2}$, $\mathcal{C}_{\alpha=2/3} = (\ell/\lambda)^{1/3}$ and $\mathcal{C}_{\alpha=1} = (\ell/\lambda)^0 = 1$, for the random walk $\alpha = 1/2$ case, the holography $\alpha = 2/3$ case and the "standard" $\alpha = 1$ case respectively. Note that none of the cumulative factors is linear in $(\ell/\lambda)$, i.e., $\delta\ell/\delta\lambda \neq \ell/\lambda$, the reason being that the $\delta\lambda$'s from the $\ell/\lambda$ parts in $\ell$ do not add coherently.

To underscore the importance of using the correct cumulative factor to estimate the spacetime foam effect, let us consider photons emitted simultaneously from a distant source coming towards our detector [22]. The fluctuating speed of light (as found in the last subsection) would seem to yield an energy-dependent spread in the arrival times of photons of energy $E$ given by $\delta t \sim \delta v(\ell/c^2) \sim t(E/E_P)^\alpha$, where $t = \ell/c$ is the average overall time of travel from the photon source (distance $\ell$ away). However, these results for the spread of arrival times of photons are not correct, because we have inadvertently used $\ell/\lambda \sim Et/\hbar$ as the cumulative factor instead of the correct factor $(\ell/\lambda)^{1-\alpha} \sim (Et/\hbar)^{1-\alpha}$. Using the correct cumulative factor, we get a much smaller $\delta t \sim t^{1-\alpha}t_P^\alpha \sim \delta\ell/c$ for the spread in arrival time of the photons [23], independent of energy $E$ (or photon wavelength, $\lambda$). Thus, the result is that the time-of-flight differences increase only with the $(1-\alpha)$-power of the average overall time of travel $t = \ell/c$ from the gamma ray bursts to our detector, leading to a time spread too small to be detectable (except for the uninteresting range of $\alpha$ close to 0).

## 2.3. Holographic Spacetime Foam

Let us consider a gedanken experiment to measure the distance $\ell$ between two points. One can determine the distance between these points by placing a clock at one of the points and a mirror at the other, and then measuring the time it takes for a a light signal to travel from the clock to the mirror [26]. However, the quantum uncertainties in the positions of the clock and the mirror introduce a corresponding uncertainty $\delta\ell$ in the distance measurement. Let us concentrate on the clock and denote its mass by $m$. Let us say the clock reads zero when the light signal is emitted and reads $t$ when the signal returns to the clock. Additionally, let us denote the uncertainties of the position of the clock at the two times by $\delta\ell(0)$ and $\delta\ell(t)$, respectively. Following Wigner, the spread in speed of the clock at time zero is given by the uncertainty principle as $\delta v = \delta p/m \gtrsim \hbar/(2m\delta\ell(0))$. This implies an uncertainty in the distance at time $t$ given by $\delta\ell(t) = (2\ell/c)\delta v \gtrsim (\hbar\ell)/(mc\delta\ell(0))$, where we have used $t = 2\ell/c$. Minimizing $(\delta\ell(0) + \delta\ell(t))/2$; then, if the position of the clock has a linear spread $\delta\ell$ at the time the light signal is emitted, this spread will grow to $\delta\ell + \hbar\ell(mc\delta\ell)^{-1}$ when the light signal returns to the clock, with the minimum at $\delta\ell = (\hbar\ell/mc)^{1/2}$, yielding $\delta\ell^2 \gtrsim \hbar\ell/(mc)$, a result first obtained by Salecker and Wigner. One can supplement this relation from quantum mechanics with a limit from general relativity [24,25]. By requiring the clock to tick off time fast enough so that the uncertainty in distance measurement is not greater than $\delta\ell$ and the clock is larger than its Schwarzschild radius $Gm/c^2$, we get $\delta\ell \gtrsim Gm/c^2$, the product of which with the inequality for $\delta\ell^2$ obtained above yields [24,25,27,28]

$$\delta\ell \gtrsim \ell^{1/3}\ell_P^{2/3}, \tag{3}$$

corresponding to the case of $\alpha = 2/3$ of Equation (1). One can also derive [6,7] the above $\delta\ell$ result by appealing to the holographic principle [8–11], which states that the maximum number of degrees of freedom $I$ that can be put into a region of space is given by the area of the region in Planck units. Accordingly, for a sphere of radius $\ell$, the entropy $S$ is bounded by $S/k_B \lesssim \frac{1}{4}4\pi\ell^2/\ell_P^2$, where $k_B$ is the Boltzmann constant. Recalling that $e^{S/k_B} = 2^I$, we get

$I \lesssim \pi(\ell/\ell_P)^2/\ln 2$. The average separation between neighboring degrees of freedom then yields $\delta\ell \gtrsim \ell^{1/3}\ell_P^{2/3}$, up to a multiplicative factor of order 1. Now it is abundantly clear that the spacetime foam model parameterized by $\alpha = 2/3$ is consistent with the holographic principle in quantum gravity, and hence it is called the holographic foam model.

### 3. Using the Most Distant Extragalactic Sources to Probe Spacetime Foam

*3.1. Spacetime Foam-Induced Phase Scrambling of Light*

Let us consider the possibility of using spacetime foam-induced phase incoherence of light from distant galaxies and gamma-ray bursts to probe Planck-scale physics. If the distance between the extragalactic source and the telescope is $\ell$, then the cumulative statistical phase dispersion is given by

$$\Delta\phi \simeq \frac{2\pi\delta\ell}{\lambda} \simeq 2\pi\frac{\ell^{1-\alpha}\ell_P^\alpha}{\lambda} \qquad (4)$$

Now to rule out models with $\alpha < 1$, one strategy [29] is to to look for interference fringes for which the phase coherence of light from the distant sources should have been lost (i.e., $\Delta\phi \gtrsim 2\pi$) for that value of $\alpha$ according to theoretical calculations.

The work above makes clear that the expected blurring of distant images is *not* merely the result of a random walk of small angle photon scatterings along the line of sight. This is because the uncertainties in the derived directions of the local wave vectors must result in the same spatial uncertainty, $\delta\ell$, regardless of how many wave crests pass the observer's location. For example, in the "thin screen approximation", the accumulated transverse path of multiply scattered photons would be approximated as $(\delta\psi)\ell >> \delta\ell$, where $\delta\psi = \Delta\phi/(2\pi)$, representing fluctuations in the direction of the local wave-vector. This would lead to expected time lags, $\delta\psi(\ell/c) >> \delta\ell/c$, in conflict with the basic premises for spacetime foam models.

This discussion brings up a thorny topic: which distance measure to use. At first glance, it would seem that the luminosity distance $D_L$ is correct, since one is measuring the path of photons. However, the accumulation of these phase differences is a function *not* of the path, but rather *of spacetime itself*. Put another way, while it is true that the luminosity distance to a source measures the distance corresponding to a flux $F = L/(4\pi D_L^2)$, where $L$ is the source luminosity, the spread in the distance traveled by the ensemble of photons $\delta\ell$ is a function of the metric of spacetime along the path. For that reason, the appropriate distance measure to use is the line-of-sight comoving distance $D_C$, given by

$$D_C(z) = D_H \int_0^z \frac{dz'}{\sqrt{\Omega_M(1+z')^3 + \Omega_k(1+z')^2 + \Omega_\Lambda}}. \qquad (5)$$

Here $D_H = c/H_0$ is the Hubble distance, and $\Omega_M, \Omega_k$ and $\Omega_\Lambda$ are the current (fractional) density parameters due to matter, curvature and the cosmological constant, respectively. Consistent with the latest WMAP and Planck data [30,31], we will use $\Omega_M = 0.27, \Omega_\Lambda = 0.73$ and $\Omega_k = 0$ and a Hubble distance $D_H = 1.3 \times 10^{26}$ m.

The fluctuations in the phase shifts over the entrance aperture of a telescope or interferometer are described by $\Delta\phi(x,y) \simeq (2\pi\ell^{1-\alpha}\ell_P^\alpha)/\lambda$, where $\{x,y\}$ are coordinates within the aperture at any time $t$. Given that the Planck scale is extremely small, we envision that $\Delta\phi(x,y)$ can be described by a random field with rms scatter $\delta\phi$ from Equation (3) without specifying whether (and if so, on what scale) in the $x - y$ plane these phase distortions are correlated. It is simplest to assume that the fluctuations are uncorrelated with one another, even down to the smallest size scale. To visualize what images of distant sources might look like after propagating to Earth through an effective "phase screen" (due to spacetime foam), consider an idealized telescope of aperture $D$. In that case, the image is just the absolute square of the Fourier transform of $e^{i\Delta\phi(x,y)}$ over the coordinates $\{x,y\}$ of the entrance aperture.

As long as $\delta\phi_{rms} \lesssim 0.6$ radians, or $\delta\ell_{rms} \lesssim 0.1\lambda$, then the Strehl ratio, which measures the ratio of the peak in the point spread function ("PSF") compared to the ideal PSF for the same optics, is approximately

$$S \simeq e^{-\delta\phi_{rms}^2}. \tag{6}$$

In addition, if these phase shifts are distributed randomly over the aperture (unlike the case of phase shifts associated with well-known aberrations, such as coma and astigmatism), then the *shape* of the PSF, after the inclusion of the phase shifts due to the spacetime foam is basically unchanged, except for a progressive decrease in $S$ with increasing $\delta\phi_{rms}$. This was demonstrated by [32], using numerical simulations of random fields. What was found, instead, was that as the variance of the random fields increased, the PSF shape *was self-similar*, aside from the appearance of the noise plateau. This contradicts the expectation from previous work (e.g., [4,13,23,29,33]) that phase fluctuations could broaden the apparent shape of a telescope's PSF.

Consider the image [29] of the active galaxy PKS1413+135 ($\ell = 1.216$ Gpc) by the Hubble Space Telescope at $\lambda = 1.6$ μm wavelength [34]. The ringlike structures around the core are clearly examples of airy rings in a diffraction-limited image, since their radii scale with the ratio $\lambda/D$, where $D$ is the telescope aperture, which would not be the case if, for example, the rings were the result of gravitational lensing. For this example, we get $\Delta\phi \sim 10 \times 2\pi$ for the random walk $\alpha = 1/2$ model and $\Delta\phi \sim 10^{-9} \times 2\pi$ for the holography $\alpha = 2/3$ model. The observation of airy rings in this case would seem to marginally rule out the random walk model. On the other hand, the holography model is obviously not ruled out. (Parenthetically, the authors of reference [29] did not reach this conclusion, as they overestimated the cumulative effects of spacetime fluctuations by a factor of $(\ell/\lambda)^\alpha$.)

### 3.2. Looking for Halo Structures

The fluctuations of spacetime foam would be directly correlated with phase fluctuations of the wavefront arriving from a distant source. Therefore, if we could observe the effects of those fluctuations, we could even more stringenty test spacetime foam models. This can be done by taking into account the expected scattering from the fluctuations. Consider a two-element interferometer observing an incoming electromagnetic wave whose local wave vector makes an angle $\theta$ with respect to the normal to the interferometer baseline. (See reference [13] for more details). The wave front develops tiny corrugations due to fluctuations in phase velocity. This causes the wave vector to acquire a cumulative, random uncertainty in direction with an angular spread $\sim \Delta\phi/(2\pi)$. Thus, it has uncertainty in both its phase and wave vector direction. Over a baseline length $D$, the magnitude of the correlated electric field would be given by

$$|E| \simeq 2E_0 \left| \cos\left( \frac{\pi}{2}[2\theta - \Delta\phi/2\pi]\frac{D}{\lambda} \right) \right|. \tag{7}$$

Working out the associated Michelson fringe visibility, we [13] find that there would be a strong reduction in visibility when $\Delta\phi \sim 2\pi\lambda/D$. Thus, by considering these changes in direction (in effect a scattering), models of spacetime foam may be tested for values of $\Delta\phi$ orders of magnitude less than possible based on the uncertainty in phase alone.

Again consider the case of PKS1413+135. With HST, which has $D = 2.4$ m, halos could be detected if $\Delta\phi \sim 10^{-6} \times 2\pi$ (as compared to $\Delta\phi \sim 10^{-9} \times 2\pi$). Thus, the HST image only fails to test the holographic model by $\sim 3$ orders of magnitude, as compared to the nine orders of magnitude discussed above for phase scrambling. However, the observation of an airy ring and the lack of a halo structure in the HST image of PKS1413+135 [34] rule out convincingly all spacetime foam models with $\alpha \lesssim 0.6$. This point allows interferometers to have a strong effect constraining spacetime foam models, an issue we discuss in Section 5.

## 4. Using Astronomical Image Archives across the Electromagnetic Spectrum

With the above work, we can invert Equation (4) to set a generic constraint on $\alpha$ for distances, $\ell$, to remote objects as a function of observing wavelength. To do this, we require $\delta\phi_{rms} = 2$ radians, thereby imposing a requirement that the Strehl ratio is ~2% of its full value. The resulting constraint is

$$\alpha > \frac{\ln(\pi\ell/\lambda)}{\ln(\ell/\ell_P)}. \tag{8}$$

The resulting constraint on $\alpha$ is shown in Figure 1. It is set simply by the detection of distant objects, rather than by observing some halo structure. As can be seen, the important result is that the observing wavelength affects very strongly the constraint one can set on $\alpha$—the shorter the wavelength, the more stringent the constraint. Thus, with observations in X-rays and $\gamma$-rays, we can set the strongest possible constraints on $\alpha$. Note that this constraint is irrespective of the detection of halo or airy ring structures, so it gives a less stringent constraint for the same wavelength than the one described in Section 3.2.

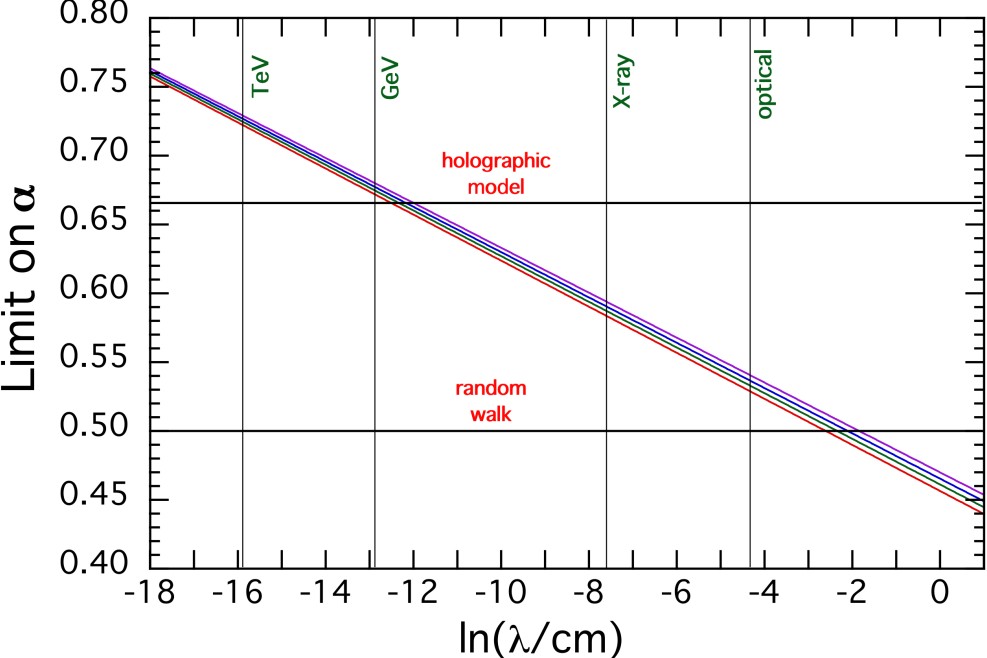

**Figure 1.** Constraints on the parameter $\alpha$, for four different comoving distances to a distant source, 425 Mpc ($z \approx 0.1$; red curve), 1 Gpc ($z \approx 0.25$; green), 3 Gpc ($z \approx 1$; blue) and 10 Gpc ($z \approx 12$; purple). The vertical dashed lines represent the optical (5000 Å), X-ray (5 keV), GeV and TeV wavebands. Since images of cosmologically distant sources betray no evidence of phase fluctuations that might be introduced by spacetime foam, there is a large region of parameter space excluded by the obeservations, denoted by the regions below the curves.

Equivalently, the $\alpha$-models predict that at any wavelengh, spacetime foam sets a maximum distance beyond which a source could not be detected because its light would be so badly out of phase that forming an image would not be possible. This is shown in Figure 2, which shows the relative flux density $\nu F_\nu$ of a source as a function of wavelength. Figure 2 and Equation (4) also show that the observing wavelength is far more powerful than distance in constraining $\alpha$, as the RMS phase shifts are $\propto \lambda^{-1}$ for any distance $\ell$.

The constraints that can be produced are summarized in Table 1. These represent lower limits to $\alpha$ produced by the mere observation of an image (not a diffraction limited one!) of a cosmologically distant source in that waveband. Thus, by far the most powerful constraints from this direction that can be put on $\alpha$ come from observations in the GeV and TeV gamma-rays. As Equation (4) makes clear, for $E_\gamma \gtrsim 1$ GeV ($\lambda <\sim 10^{-15}$ m) the

wavelengths are sufficiently short that the phase shifts can exceed $\pi$ radians for $\alpha \simeq 2/3$. Merely detecting a well-localized image of a cosmologically distant object can rule out the holographic model and place serious constraints on $\alpha$. This would appear to rule out the holographic model. However, for reasons to be discussed in Section 6, we believe it is still possible to reconcile this result with the holographic principle.

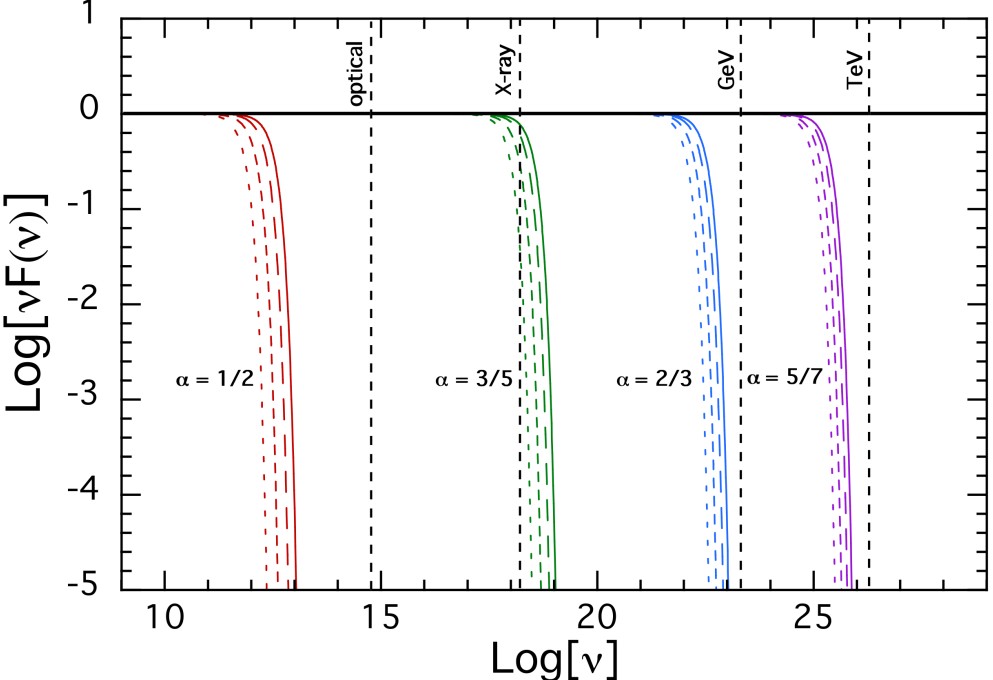

**Figure 2.** The relative observed flux density $\nu F_\nu$, as a function of frequency $\nu$. Curves are shown for comoving distances of 425 Mpc ($z \approx 0.1$; solid), 1 Gpc ($z \approx 0.25$; long-dashed), 3 Gpc ($z \approx 1$; medium-dashed) and 10 Gpc ($z \approx 12$; short-dashed). The plotted curves assume $\alpha =1/2$ (red), 3/5 (green), 2/3 (blue) and 5/7 (purple). They imply that for a given value of $\alpha$, there will be a maximum frequency $\nu$ (or equivalently, a shortest wavelength $\lambda$) beyond which a source would simply be undetectable, as the ensemble of photons received from the source would have a phase dispersion of $\Delta\phi \sim 1$ radian, making an image impossible to form.

　　Gamma-ray telescopes rely on detecting the cascades of interactions with normal matter, be it crystal scintillators (as used in the GeV band) or the Earth's atmosphere (in the TeV band). Cosmologically distant sources have in fact been detected in both the GeV and TeV bands, as demonstrated by [32] and references therein. In particular, at GeV energies, *Fermi* has obtained images for well over three thousand AGN [35], of which over 98% are blazars. A large number of the sources in that catalog are at redshifts over $z = 3$.

　　At TeV energies, the highest redshift source is at $z = 1.1$ [36]. This upper envelope is produced by photons in the optical-IR extragalactic background light (EBL, [37]), which interact with very high-energy gamma-rays to produce electron–positron pairs. These photons can then cascade to produce lower-energy photons. A detailed study of the EBL [38] shows that the optical depth $\tau$ due to pair production becomes non-negligible ($\tau < 0.1$) at around 50–100 GeV for sources at $z = 0.5$–1 and at around 300 GeV for sources at $z = 0.1$. However, because the effect we address here regards the mere formation of an image, the appropriate threshold is $\tau \approx 1$–2. Optical depth $\tau = 1$ is reached at $\sim$1 TeV for sources at $z = 0.1$ (red line in Figure 1, solid curves in Figure 2) and 100–200 GeV for sources at $z = 0.5$–1 (see [38], their Figure 7). For $\tau > 1$, the majority of the photons observed can be secondaries from the resulting cascades (which originate a tiny distance from the pair-production event, [39]), rather than primaries from the source itself. As a result, it is possible for a source, if detected, to form an image at distances much larger than $\ell = 425$ Mpc. We note that while the cascading photons can form a halo to the image of

the source at TeV photon energies due to the influence of the intergalactic magnetic field (IGMF, see [40], particularly their Figure 6, which simulates the image of a $z \simeq 0.14$ BL Lac object for a choice of IGMF), that is secondary to whether an image is formed at all.

As a result, the TeV $\gamma$-ray result in Table 1 reflects a path length of $\ell = 425$ Mpc ($z = 0.1$). By comparison, at photon energies below 100 GeV, we are able to use the highest redshift sources that have been observed in those wavebands. All told, the existing observations place a limit of $\alpha > 0.72$ on the phenomenology of spacetime foam models.

**Table 1.** Constraints on the spacetime foam parameter $\alpha$.

| Waveband | Lower Limit [a] on $\alpha$ |
|---|---|
| optical (eV) | 0.53 |
| X-ray (keV) | 0.58 |
| $\gamma$-rays (GeV) | 0.67 |
| $\gamma$-rays (TeV) | 0.72 |

[a] See Equation (8) and Figure 1.

### 5. Other Results and Future Prospects

The discussion in Section 3.2 makes it possible to further test the phenomenology of quantum foam with interferometers. Interferometric arrays rely on the ability to achieve phase coherence in the wavefront to image distant sources. Here, we do not propose to use them as imagers, but rather as *spatial filters*. Namely, we can use Equation (7), which predicts that there will be a strong reduction in visibility when $\Delta\phi \sim 2\pi\lambda/D$. This was first proposed by [13], who pointed out that using this method, imaging from the VLTI would begin to probe the $\alpha = 2/3$ model. However, in the light of the results of [32], presented in Section 4, there is already persuasive evidence against that model. That is further strengthened by the observations of 3C 273 (which at $z = 0.158$ has a path length $\ell \approx 700$ Mpc) with the *GRAVITY* instrument [41], which shows the interferometric detection of that quasar's spectrum in its Pa $\alpha$ line at about 2 $\mu$m. The detection of fringes from this source on a 140 m baseline also rules out the $\alpha = 2/3$ model.

As Section 3.2 notes, wavelength is a critical factor in the ability of a given observation to provide evidence against a given spacetime foam model. Optical observations are at a wavelength that is a factor $\sim$5 shorter, so detection of fringes with the Magdalena Ridge Optical Interferometer (maximum baseline length $D = 340$ m) in the $B$ band from PKS 1413+135 requires at most $\Delta\phi \sim 8.4 \times 10^{-9}$, as compared to the prediction of $\Delta\phi = 9 \times 10^{-8}$ for PKS 1413+135. Thus, the observation of fringes from distant quasars from the MROI and/or VLTI in the optical will be able to set limits on $\alpha$ similar to those discussed in Section 4 (see also Table 1). However, X-ray interferometry would be able to make a much larger impact. Such a mission concept has been proposed for the ESA *Vision 2050* Programme [42], but it is not at all clear that the mission will be approved and/or technically feasible, as the technology challenges are considerable.

### 6. Conclusions

We have discussed the small-scale fluctuations that may be introduced by spacetime foam in the wavefronts of cosmologically distant objects, if the fluctuations add up at all. In particular, we have discussed three possible models, the random walk and holography models, where the fluctuations do add up, and a third model, the "standard" case, corresponding to the original Wheeler [1–3] conjecture, where they do not. Astronomical observations of distant AGN in the optical and all shorter wavelengths, decisively rule out the random walk model. Astronomical observations of distant AGN in the TeV $\gamma$-rays strongly disfavor, and may altogether rule out, the holography model. A similar statement can be said regarding the observation of interferometric fringes from 3C 273 with the VLTI.

Here we must be careful what the available data *do* and *do not* rule out. What they *do* is rule out values of the accumulation power $\alpha$ that are lower than 0.72. However, they *do not*

necessarily rule out the application of the holographic principle to spacetime foam models, since that model can take on a different and more subtle form than that encapsulated by Equation (2). As pointed out in Section 2, the main assumption used to derive the $\alpha = 2/3$ model from quantum mechanics was a requirement that the mass and size of the system under consideration satisfy $M < \ell c^2/2G$ so that outside observations are actually possible. However, as discussed in [32], this can be waived if gravitational collapse does not necessarily produce an event horizon, as proposed by [43,44].

There is a second caveat, which could be more important. In this work we have considered instantaneous fluctuations in the distance between a distant source and a given point on the telescope aperture. It is possible that one needs to average over the huge number of Planck timescales that it takes for light to propagate through a telescope, or the similarly large number of Planck squares across a telescope's aperture. This could make the fluctuations we have been discussing vanish. However, at the moment we do not even have a formalism for carrying out such averages.

Another, interesting avenue is the proposal [6,16] to use gravitational-wave interferometers to detect spacetime foam. The idea is that uncertainties in distance measurements would manifest themselves as a displacement noise that infests the interferometer. It has been suggested that modern gravitational-wave interferometers appear to be within striking distance of testing the holographic model. Unfortunately, those studies have concentrated on the observation along the propagation direction of light in the interferometer. While those are important, the beam size in the transverse direction is a matter of concern, because the discussion above assumes implicitly that spacetime in between the mirrors in the interferometer fluctuates coherently for all photons in the beam. However, the large beam size in LIGO makes such coherence unlikely [4]. This coherence problem might be mitigated with the use of the twin table-top 3D interferometers [45] under construction at Cardiff University in Wales.

**Author Contributions:** Conceptualization, E.S.P. and Y.J.N.; Methodology, E.S.P. and Y.J.N.; Investigation, E.S.P. and Y.J.N.; writing—original draft preparation, E.S.P. and Y.J.N.; writing, review and editing—E.S.P. and Y.J.N. All authors have read and agreed to the manuscript.

**Funding:** This research was supported in part by the US Department of Energy, the Bahnson Fund and the Kenan Professors Research Fund of the University of North Carolina.

**Institutional Review Board Statement:** Not applicable.

**Informed Consent Statement:** Not applicable.

**Data Availability Statement:** Not applicable.

**Acknowledgments:** This review paper is based on work done with our collaborators W. Christiansen, the late H. van Dam, S.A. Rappaport, J. DeVore, D. J. E. Floyd and D. Pooley. We thank all of them for stimulating discussions and fruitful collaborations. Y.J.N. is also grateful to G. Amelino-Camelia and R. Weiss for useful conversations. E.S.P. is grateful to J. A. Franson, F. W. Stecker and M. Livio for interesting discussions.

**Conflicts of Interest:** The authors declare no conflict of interest.

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
