# Peer review of "Probing Spacetime Foam with Extragalactic Sources of High-Energy Photons"

_universe, doi:10.3390/universe8070382_

Round 1

Reviewer 1 Report

This paper presents an interesting review on some results about the possibility to detect the supposed quantum structure of spacetime using the detection of electromagnetic radiation originated by extragalactic sources. In this work are proposed some tests on how to detect the quantum foam spacetime observing the photon decoherence caused by the propagation in a quantized space. This paper is suited for publication in Symmetry after some minor revisions.

The first improvement can be made in section 2.3, here the formula of the improved clock position uncertainty can be better introduced (line 117). It is clear the meaning, but the authors can explain how the formula is obtained introducing an uncertainty related to the wavelenght.

Unother improvement can be made in section 3.1, the sentence in line 165-167 "However,..." cen be better explained.

In line 177 the word "rms" must be a typo.

Perhaps it could be better to underline that in section 4 the experimental data suggest that the holographic model can be ruled out, but under some assumptions that are explained in the conclusion the holographic principle can still be valid.

In the introduction the authors use the reference [13] to justify the possibility to use UHECR to test the supposed quantum structure of spacetime. Perhaps it could be better to cite the papers: arXiv:0811.2230, arXiv:2110.09900, arXiv:2110.09184 as well. In DSR theories such as in the reference [13] the quantum gravity modifications feeled by propagating particles are universal and it poses a strenght constraint on the possibility to modify the threshold energies required in the GZK phenomenon. In other quantum gravity scenarios such as SME, the modifications are supposed not universal and the GZK threshold energy can be modified.

As a final remark this work is interesting and suited for publication, but requires the suggested minor revisions.

Best Regards.

Author Response

We thank the referee for his/her useful comments.  Below, we detail our responses and comments.  Where we have made changes, they are boldfaced.

We thank the referee for calling our attention to three additional references regarding UHECR.  Accordingly we have added them to the paper.

The referee’s question about the derivation of Equation (3) is, we believe, related to some steps we have skipped in the derivation.  As we noted, the gedanken experiment we use was first proposed by Wigner, and our derivation is depends on 2 separate (consecutive) arguments: the first one depending on quantum mechanics (Heisenberg’s uncertainty) which was proposed by Wigner, and the second one depending on black hole physics which H. van Dam and one of us (Ng) used. While we are not quite sure what is unclear, we have bolstered this section by eliminating the missing steps in the part that uses quantum mechanics and making the derivation more explicit.  Please see lines 113-119.

We have added a sentence better justifying the use of the comoving distance rather than the luminosity distance (lines 157-160). Again, the changes are boldfaced.

Regarding the suggestion to rephrase the conclusions from Section 4 to state that the holographic principle can be rescued due to points raised in the conclusion. I have added two sentences to the end of Section 4 (formerly lines 240-242, now lines 246-248), clarifying this point.  We have also removed some awkward phrasing and typographical errors in lines 234-5 and 238.

Reviewer 2 Report

In "Probing spacetime foam with extragalactic sources of high-energy photons" the authors explore the idea that high-energy photons can be used to probe the spacetime foam. This work is important for many reasons, and takes advantage of the many observations of electromagnetic radiation across the whole spectrum.

Overall, I find the article well-written and the ideas interesting. I am missing, however, a more thorough discussion on the possible uncertainties of the analyses. I also think some parts could be expanded to give the reader a better idea of the analysis. This is particularly important considering that this article dialogues with two communities, both those studying the space-time foam, and astrophysicists.
Below I list some concrete examples and other general points. The one marked '*' is the one that actually could majorly affect the results.

- In equation 4 a phase is introduced. This observable, while essential to probe the space-time foam, depend on other properties of the medium. One example is Faraday rotation, which is very important at low energies (radio frequencies) but whose impact at higher energies I can't easily tell.

- Another effect that has to be properly identified is vacuum birefringence. This is most likely to happen, although it may ultimately be completely negligible.

- I believe I understood the motivation for adopting the comoving instead of the luminosity distance, as defined in equation 5. However, I'm not completely sure of it, and I think a non-expert reader might need more convincing arguments for this.

- The formation of rings, as described in section 3, is something that may also happen due to purely gravitational effects. How to separate from a putative signal of the space-time foam?

- I encourage the authors to define what they mean by "high-energy photons" and "ultra-high-energy". In particular, a common nomenclature confusion in the (sub-)fields is the use of "ultra-high-energy" to describe photons with E > 100 TeV, when there has been for a long time a whole sub-field focused on studying ultra-high-energy cosmic rays (often defined as particles with E > 1 EeV).

- Around equation 3, I am curious about a possible role of curvature around large objects. Does it affect this quantity? Is the metric preserved? If not, can there be non-diagonal metric terms?

* My main concern relates to the gamma-ray analysis. One cannot perform an analysis at this energy band considering cosmologically distant source without including propagation effects in the analysis. The main propagation effects are related to the development of electromagnetic cascades in the intergalactic medium. Above ~300 GeV gamma rays produce electron-positron pairs by interacting with the extragalactic background light (EBL), which quickly (after ~10-100 kpc) scatter background photons (mostly CMB) to high energies, creating a cascade. The results presented in the paper require primary photons, i.e., those that were not produced in the cascade, which at TeV energies are not many.
The development of the cascade depend on the presence of intergalactic magnetic fields (IGMFs) (see, e.g., 2010.10525 and 2105.12020 and references therein for details on this topic).

- Continuing the previous point, the electron-positron pairs composing the cascade can be deflected by IGMFs. This may cause time delays that can affect GeV-TeV photons and make it hard to perform time-dependent analysis along the lines of what was proposed in the manuscript.

- As a final remark, measuring phases at high energies is difficult. There are very few proposals for gamma-ray polarimetry. I encourage the authors to stress the importance of such measurements for fundamental physics studies, like the one proposed.

In summary, after carefully reading the manuscript, I cannot recommend it for publication, at this stage, for the reason pointed in '*'. If this particular issue is clarified, I would be happy to reconsider my recommendation. I also identified some points that could help improve the manuscript.

Author Response

We thank the referee for his/her useful comments.  Below, we detail our responses and comments.  Where we have made changes, they are boldfaced.

The referee is correct that TeV gamma-rays interact with the EBL, producing electromagnetic cascades, and that this has been used to place limits on intergalactic magnetic fields. In particular, he/she claims that primary photons, that were produced in the source, not as a result of the cascades, are few at TeV energies, making the entire method infeasible because the electron-positron pairs can be deflected by the IGMF.   This conclusion cannot be supported by the evidence.  The simulations shown in those papers are highly model dependent, and only serve to put an upper limit on magnetic fields.   In any case, good reviews regarding these cascades and the attenuation of the blazar emission due to them as a function of redshift can be found in Abeyskara et al. 2019, ApJ, 885, 150, as well as Dominguez et al. 2019, ApJ, 885, 137.  It is important to note that neither of those papers, nor those referenced by them, hold that the photons detected in the TeV are secondary in nature, i.e., before detection they have gone through a pair-production cascade and then the resultant particles inverse-Comptonize CMB radiation. 

To the contrary,  while the fraction of photons detected from the source (compared to those emitted) decreases with increasing redshift due to the cascades, the photons detected *must be primary*.  This is for two reasons.  First of all, their energy spectrum is consistent with inverse-Compton emission with the seed photons being within the source (from the jet itself or other AGN regions such as the broad-line region).  If instead the majority of TeV photons from distant sources were secondary the seed photons would be CMB, not AGN related — and this is inconsistent with the spectral shape.  Second, flares in the TeV are invariably correlated with flares seen at lower energies (e.g., X-rays, optical, etc).  This would not be true if the TeV photons were secondary — instead one would expect that the flares would not be seen at all because of the long propagation lengths of the cascading particles.  Good discussions of such work can be found in, for example Adams et al. 2022 (ApJ, 924, 95),  Pandey, Gupta & Wiita 2018 (ApJ, 859, 49); Ding et al., 2017, MNRAS, 464, 599; Abdalla et al. 2017, A&A, 600, A89; Zhang et al. 2012, ApJ, 752, 157 and many other papers. For these reasons we have made no changes to the paper regarding the use of the gamma-ray tests.

We have clarified in the abstract what we mean by ultra-high-energy photons, following the use in the TeV

astrophysics community (line 6).  We appreciate the referee’s encouragement to clarify this point, as this is a rather different usage than the particle physics one.  The change is boldfaced.

No modification is necessary relative to the issue of Faraday rotation, as it is proportional to $1/\nu^2$, where $\nu$ is the frequency of the light.  Therefore, while Faraday rotation is relevant in the radio, it is irrelevant in optical, X-ray or Gamma-ray astronomical observations.

Similarly, since the spread in $\ell$ is a function of the spacetime metric it could well change near large objects.  However those would be the same objects where gravitational lensing is present. Such objects are rare and of extreme interest.  But, as we emphasize in section 3.2, the fluctuations in the phase shifts (due to spacetime foam) over the entrance aperture of telescope or interferometer are distributed randomly, whereas purely gravitational effects do not give rise to such random distributions.  Moreover, typically the path length differences between different lensed images is of order light-months for galaxy-sized objects, and light-minutes to light-hours for stellar-sized objects, many orders of magnitude greater than the spreads due to spacetime foam effects.   We have, however, made our derivation of equation (3) more explicit, in response to these comments and (more critically) those made by Referee#1.  Those changes are boldfaced in lines 113-119.

Regarding the point about vacuum birefringence, evidence of vacuum birefringence was found only recently by R. P. Mignani et al. (arXiv: 1710.08709) because of the rather extremely large magnetic fields that are required. If we want to probe spacetime foam (a minute result requiring cumulative effects) in a (local) medium exhibiting vacuum birefringence, the difficulties would be exponentially increased.  We agree with the referee that the effect would be more or less completely negligible. 

Round 2

Reviewer 2 Report

I thank the the authors for taking my comments into consideration.
The main issue I raised, unfortunately, has not been addressed convincingly (but all the other were).

--> "The referee is correct that TeV gamma-rays interact with the EBL, producing electromagnetic cascades, and that this has been used to place limits on intergalactic magnetic fields. In particular, he/she claims that primary photons, that were produced in the source, not as a result of the cascades, are few at TeV energies, making the entire method infeasible because the electron-positron pairs can be deflected by the IGMF. This conclusion cannot be supported by the evidence."

To be precise, my claim is that primary photons *may* be the dominant ones (I'm not affirming it). The method is still useful if: 1) IGMFs are taken into account; 2) They are shown to lead to smaller effects than the one you are seeking; 3) the spectrum is such that the flux of photons below the threshold for pair production is much greater than the contribution of the EBL-reprocessed secondary photons. This is my first criticism and this very same argument holds regardless of IGMFs. In the presence of IGMFs, one is increasing the rate of interactions with the background photon fields (EBL, CMB), thereby generating more secondaries, making the cascade effect even more important.

I would be happy to discuss why "this conclusion cannot be supported by the evidence". This is a well-known effect and I cannot see how one could argue against it. In my view, what is open for discussion is if IGMFs are weak or even 0, or if some other effect like plasma instabilities is at play quenching the cascade. There is no room for discussion about the existence of the cascade effect.  It is undeniable that there is pair production, and to some degree, inverse Compton. These well-known phenomena might not be dominant and might be irrelevant for your analysis, but the burden of proof falls on you.

--> The simulations shown in those papers are highly model dependent, and only serve to put an upper limit on magnetic fields.

Definitely the simulations are model-dependent. But there are detailed 3D simulations covering the whole parameter space of IGMFs. There are some that are actual fits as opposed to lower limits. The review papers I suggested in the first round of review mention some of them.  
In any case, the simulations do not "only serve" to put lower (not upper) limits on the IGMFs. There is fundamental physics involved, things that are well known and that cannot be ignored when searching for the spacetime foam.

--> In any case, good reviews regarding these cascades and the attenuation of the blazar emission due to them as a function of redshift can be found in Abeyskara et al. 2019, ApJ, 885, 150, as well as Dominguez et al. 2019, ApJ, 885, 137. It is important to note that neither of those papers, nor those referenced by them, hold that the photons detected in the TeV are secondary in nature, i.e., before detection they have gone through a pair-production cascade and then the resultant particles inverse-Comptonize CMB radiation.

TeV photons can be secondary if the sources emit very energetic (>> TeV) photons. In general, my argument goes for the ~GeV-TeV band. A secondary TeV gamma ray would have to come from a very close source.
In the works mentioned they fit an exponential or log-parabola spectrum with parameters E0 (cut-off energy) and something like a spectral index. These parameters, indeed, tend to favour the interpretation that secondary photons are sub-dominant (not non-existent), especially if the primaries have energies not much greater than ~TeV and soft spectra. The fact that these works make no explicitly reference to the existing effect is because it is not of primary concern for what they are doing. For instance, Domínguez et al. only analysed data ~100 GeV, where primary gamma dominate in most cases. Even reviews on the exact topic of this manuscript (e.g., https://arxiv.org/abs/2111.05659) acknowledge the importance of the cascade treatment, even though it is in a very subtle way.
Furthermore, the references mentioned above deal with the EBL, which is an isotropic background. They perform a phenomenological analysis of the observations, with little regard for what is in-between (propagation effects). They don't mention IGMFs, which could be done, but that is not a problem because IGMFs would likely not significantly affect the shape of the spectrum anyways. (This follows from Liouville's theorem.)

--> To the contrary, while the fraction of photons detected from the source (compared to those emitted) decreases with increasing redshift due to the cascades, the photons detected *must be primary*. This is for two reasons. First of all, their energy spectrum is consistent with inverse-Compton emission with the seed photons being within the source (from the jet itself or other AGN regions such as the broad-line region).  If instead the majority of TeV photons from distant sources were secondary the seed photons would be CMB, not AGN related — and this is inconsistent with the spectral shape.

The statement that these photons "must be primary" is wrong. The correct way to put it would be "they are mostly primary". These photons can be secondary (otherwise, I would expect a proof that there are 0 secondary photons).
The physics is fairly simple. A very rough analytical treatment is provided in https://arxiv.org/abs/0910.1920. The mean free path for pair production is given by eq. 19, and for inverse Compton by eq. 28. For instance, for a 20 TeV gamma ray propagating in the intergalactic medium, its mean free path is roughly 40 Mpc. It generates pairs that take, say, half of the gamma-ray energy, so 10 TeV each (it's not exactly that, but this is useful for discussion). Because of the inelasticity of inverse Compton scattering, most of these 10 TeV will be transferred to photons that will reach only a fraction of this energy, probably less than 100 GeV. Therefore, I do expect sources of ~10 TeV gammas to contribute in the ~1-100 GeV energy range with secondary photons.

After some literature digging, I found two plots that show how much of the flux from primary TeV source is in reality secondary: https://arxiv.org/abs/2009.09772 (fig. 1) and https://arxiv.org/abs/1701.00654 (fig. 7). These are based on Monte Carlo simulations. There is some room for changes due to EBL uncertainties, but this would only change things by a factor of a few (even if it is a factor ~10, the argument holds). It shows, uncontroversially, that secondary photons may dominate the 1-100 GeV band.

There seems to be some confusion with respect to what exactly is observed. One thing is what happens in the source, the other is what I'm saying about propagation. The spectrum measured at Earth is a combination of both, and it is in principle arbitrary. Moreover, it can be degenerate (source spectrum * propagation).
Whether the seed photons are the CMB or AGN, it only matters where the energy comes from (in this case AGN). I don't see why the spectral shape would inform us *unambiguously* on the origin of the photons if other processes are at play.

--> Second, flares in the TeV are invariably correlated with flares seen at lower energies (e.g., X-rays, optical, etc). This would not be true if the TeV photons were secondary — instead one would expect that the flares would not be seen at all because of the long propagation lengths of the cascading particles. Good discussions of such work can be found in, for example Adams et al. 2022 (ApJ, 924, 95), Pandey, Gupta & Wiita 2018 (ApJ, 859, 49); Ding et al., 2017, MNRAS, 464, 599; Abdalla et al. 2017, A&A, 600, A89; Zhang et al. 2012, ApJ, 752, 157 and many other papers. For these reasons we have made no changes to the paper regarding the use of the gamma-ray tests.

That is correct, the correlation exists. But the point is: X-rays, MeV gamma rays, optical, radio, etc, they are all primary (maybe with some contribution from the background which is irrelevant for the discussion). Everything is primary until the energy of the photon is such that it exceeds the kinematic threshold for pair production with some cosmological radiation field (in this case, the EBL). Then cascade occurs. This threshold is roughly ~300 GeV. Therefore, if a source emits ~10 GeV gamma rays and they arrive at Earth, they are *certainly* primaries. However, if you observer a ~10 GeV gamma ray, you cannot be sure if it is primary or secondary.

The main problem would be the range between GeV and TeV, where secondary photons may dominate. In any case, the observation of a correlation does not exclude the existence of another underlying process which may be suppressing part of the flux. Even if a flux is observed, there is no guarantee that it wasn't originally much larger. Therefore, the correlation of TeV flares with other wavelengths proves absolutely nothing in this context.

The references mentioned are dealing with the physics in the source, not at propagation. Most of them mimic the propagation effects afterwards by simply taking an energy-dependent exponential cut-off with exponent equal to the optical depth of the EBL. This is a good approximation in most cases (if IGMFs are ignored). But again, they are not looking for tiny effects as is the case of this work.

The authors say that "instead one would expect that the flares would not be seen at all because of the long propagation lengths of the cascading particles". This depends on the properties of IGMFs. Take a look at the time delay in the simple analytical estimates of https://arxiv.org/abs/0910.1920. For weak magnetic fields (< 1e-20 G), things would look pretty much the same; but there is a transition region (B > 1e-15 G) where time delays might be important. These are actually in line with the existing constraints by Fermi-LAT (https://arxiv.org/abs/1804.08035), in addition to the first evidence of strong IGMFs published in Science in 2010 (https://arxiv.org/abs/1006.3504) and many other works (see reviews in https://arxiv.org/abs/2105.12020, https://arxiv.org/abs/2010.10525, etc). Because these fields are very uncertain, it is fine, in my opinion, to proceed with the analysis by neglecting them; but their possible role ought to be acknowledged.

Another thing about the correlations is that the GeV part of the flare can be diluted with respect to the TeV part considering IGMFs. The observation of a correlation does not preclude the existence of another underlying process which may be suppressing a fraction of the total flux.

In summary, I do not find the authors' reply convincing with respect to the gamma-ray part. In fact, it appears to contain some fundamental flaws. In particular, the authors seem to question the basic principle of electromagnetic cascades (even in the absence of IGMFs). Sure, IGMFs are not well known and they can (and should) be questioned. Indeed, most analyses are not accurate enough. But if IGMFs are non-negligible, the whole method present is spoiled (or will have to be adjusted). To me it is evident that if pair production occurs on scales of ~100 Mpc (for ~10 TeV photons), and inverse Compton on ~100 kpc scales, the latter generating a bunch of secondary photons with energies as high as ~100 GeV in this case, then the secondary signal is at least in parts not primary. Moreover, if IGMFs are non-negligible, the pairs  will be affected by them, undergoing deflections (and to a much lesser degree due to synchrotron emission).  And I haven't even started to talk about other effects that can affect the cascades such as plasma instabilities.

To probe tiny effects associated with the spacetime foam, it is important to understand all other sources of small uncertainties. I'm not saying that the method does not work, but such caveats should be taken into account or at the very least discussed. To be honest, as someone who works on the field, I can say that it is not practical to include these effects at the moment. But it is completely reasonable to discuss them, in my opinion, if the authors convince themselves that the effects I'm pointing to may play a role. Otherwise the results presented in the paper (fig. 2) will be misleading.

Author Response

After detailed discussions with Floyd Stecker, an expert on the extragalactic background light and gamma-ray propagation, we have added a paragraph concerning the effects of cascades. This paragraph (boldfaced) is now the penultimate paragraph of Section 4.  We note that no change was necessary to Figure 1 or Figure 2, as those already included lines/curves for z=0.07, where sources are completely unaffected by cascades at 1 TeV,  and z=0.25, where the optical depth at 1 TeV is of order 2.   The same is true of Table 1, as the alpha=0.72 value is reached for a 420 Mpc path length. We have also added three new references (Stecker et al., Plaga , and Batista and Saveliev, the last recommended by the referee).